# Knowledge as a Teacher:
# Knowledge-Guided Structural Attention Networks

## Abstract

Natural language understanding (NLU) is a core component of a dialogue system. Recently recurrent neural networks (RNN) obtained strong results on NLU due to their superior ability of preserving sequential information over time. Traditionally, the NLU module tags semantic slots for utterances considering their flat structures, as the underlying RNN structure is a linear chain. However, natural language exhibits linguistic properties that provide rich, structured information for better understanding. This paper introduces a novel model, knowledge-guided structural attention networks (K-SAN), a generalization of RNN to additionally incorporate non-flat network topologies guided by prior knowledge. There are two characteristics: 1) important substructures can be captured from small training data, allowing the model to generalize to previously unseen test data; 2) the model automatically figures out the salient substructures that are essential to predict the semantic tags of the given sentences, so that the understanding performance can be improved. The experiments on the benchmark ATIS data show that the proposed K-SAN architecture can effectively extract salient knowledge from substructures with an attention mechanism, and outperform the state-of-the-art neural network based frameworks.

## 1 Introduction

In the past decade, goal-oriented spoken dialogue systems (SDS), such as the virtual personal assistants Microsoft's Cortana and Apple's Siri, are being incorporated in various devices and allow users to speak to systems freely in order to finish tasks more efficiently. A key component of these conversational systems is the natural language understanding (NLU) module-it refers to the targeted understanding of human speech directed at machines (Tur and De Mori, 2011). The goal of such "targeted" understanding is to convert the recognized user speech into a task-specific semantic representation of the user's intention, at each turn, that aligns with the back-end knowledge and action sources for task completion. The dialogue manager then interprets the semantics of the user's request and associated back-end results, and decides the most appropriate system action, by exploiting semantic context and user specific meta-information, such as geo-location and personal preferences (McTear, 2004; Rudnicky and Xu, 1999).

A typical pipeline of NLU includes: domain classification, intent determination, and slot filling (Tur and De Mori, 2011). NLU first decides the domain of user's request given the input utterance, and based on the domain, predicts the intent and fills associated slots corresponding to a domain-specific semantic template. For example, Figure 1 shows a user utterance, "*show me the flights from seattle to san francisco*" and its semantic frame, find_flight(origin="seattle", dest="san francisco"). It is easy to see the relationship between the origin city and the destination city in this example, although these do not appear next to each other. Traditionally, domain detection and intent prediction are framed as utterance classification problems, where several classifiers such as support vector machines and maximum entropy have been employed (Haffner et al., 2003; Chelba et al., 2003; Chen et al., 2014). Then slot filling is framed as a word sequence tagging task, where the IOB (in-out-begin) format is ap-

show me the flights from seattle to san francisco

↓ ↓ ↓ ↓ ↓ ↓ ↓ ↓ ↓

O O O O O B-origin O B-dest I-dest

Figure 1: An example utterance annotated with its semantic slots in the IOB format (S).

plied for representing slot tags as illustrated in Figure 1, and hidden Markov models (HMM) or conditional random fields (CRF) have been employed for slot tagging (Pieraccini et al., 1992; Wang et al., 2005).

With the advances on deep learning, deep belief networks (DBNs) with deep neural networks (DNNs) have been applied to domain and intent classification tasks (Sarikaya et al., 2011; Tur et al., 2012; Sarikaya et al., 2014). Recently, Ravuri and Stolcke (2015) proposed an RNN architecture for intent determination. For slot filling, deep learning has been viewed as a feature generator and the neural architecture can be merged with CRFs (Xu and Sarikaya, 2013). Yao et al. (2013) and Mesnil et al. (2015) later employed RNNs for sequence labeling in order to perform slot filling. However, the above studies benefit from large training data without leveraging any existing knowledge. When tagging sequences RNNs consider them as flat structures, with their underlying linear chain structures, potentially ignoring the structured information typical of natural language sequences.

Hierarchical structures and semantic relationships contain linguistic characteristics of input word sequences forming sentences, and such information may help interpret their meaning. Furthermore, prior knowledge would help in the tagging of sequences, especially when dealing with previously unseen sequences (Tur et al., 2010; Deoras and Sarikaya, 2013). Prior work exploited external web-scale knowledge graphs such as Freebase and Wikipedia for improving NLU (Heck et al., 2013; Ma et al., 2015b; Chen et al., 2014) Liu et al. (2013) and Chen et al. (2015) proposed approaches that leverage linguistic knowledge encoded in parse trees for language understanding, where the extracted syntactic structural features and semantic dependency features enhance inference model learning, and the model achieves better language understanding performance in various domains.

Even with the emerging paradigm of integrating deep learning and linguistic knowledge for different NLP tasks (Socher et al., 2014), most of the previous work utilized such linguistic knowledge and knowledge bases as additional features as input to neural networks, and then learned the models for tagging sequences. These feature enrichment based approaches have some possible limitations: 1) poor generalization and 2) error propagation. Poor generalization comes from the mismatch between knowledge bases and the input data, and then the incorrectly extracted features due to errors in previous processing propagate errors to the neural models. In order to address the issues and better learn the sequence tagging models, this paper proposes knowledge-guided structural attention networks, K-SAN, a generalization of RNNs that automatically learn the attention guided by external or prior knowledge and generate sentence-based representations specifically for modeling sequence tagging. The main difference between K-SAN and previous approaches is that knowledge plays the role of a teacher to guide networks where and how much to focus attention considering the whole linguistic structure simultaneously. Our main contributions are three-fold:

- End-to-end learning
  To our knowledge, this is the first neural network approach that utilizes general knowledge as guidance in an end-to-end fashion, where the model automatically learns important substructures with an attention mechanism.
- Generalization for different knowledge
  There is no required schema of knowledge, and different types of parsing results, such as dependency relations, knowledge graph-specific relations, and parsing output of handcrafted grammars, can serve as the knowledge guidance in this model.
- Efficiency and parallelizability
  Because the substructures from the input utterance are modeled separately, modeling time may not increase linearly with respect to the number of words in the input sentence.

In the following sections, we empirically show the benefit of K-SAN on the targeted NLU task.

## 2 Related Work

**Knowledge-Based Representations** There is an emerging trend of learning representations at different levels, such as word embeddings (Mikolov et al., 2013), character

embeddings (Ling et al., 2015), and sentence embeddings (Le and Mikolov, 2014; Huang et al., 2013). In addition to fully unsupervised embedding learning, knowledge bases have been widely utilized to learn entity embeddings with specific functions or relations (Celikyilmaz and Hakkani-Tur, 2015; Yang et al., 2014). Different from prior work, this paper focuses on learning composable substructure embeddings that are informative for understanding.

Recently linguistic structures are taken into account in the deep learning framework. Ma et al. (2015a) and Tai et al. (2015) both proposed dependency-based approaches to combine deep learning and linguistic structures, where the model used tree-based n-grams instead of surface ones to capture knowledge-guided relations for sentence modeling and classification. Roth and Lapata (2016) utilized lexicalized dependency paths to learn embedding representations for semantic role labeling. However, the performance of these approaches highly depends on the quality of "whole" sentence parsing, and there is no control of degree of attentions on different substructures. Learning robust representations incorporating whole structures still remains unsolved. In this paper, we address the limitation by proposing K-SAN to learn robust representations of whole sentences, where the whole representation is composed of the salient substructures in order to avoid error propagation.

**Neural Attention and Memory Model** One of the earliest work with a memory component applied to language processing is memory networks (Weston et al., 2015; Sukhbaatar et al., 2015), which encode facts into vectors and store them in the memory for question answering (QA). Following their success, Xiong et al. (2016) proposed dynamic memory networks (DMN) to additionally capture position and temporality of transitive reasoning steps for different QA tasks. The idea is to encode important knowledge and store it into memory for future usage with attention mechanisms. Attention mechanisms allow neural network models to selectively pay attention to specific parts. There are also various tasks showing the effectiveness of attention mechanisms.

However, most previous work focused on the classification or prediction tasks (predicting a single word given a question), and there are few studies for NLU tasks (slot tagging). Based on the

fact that the linguistic or knowledge-based substructures can be treated as prior knowledge to benefit language understanding, this work borrows the idea from memory models to improve NLU. Unlike the prior NLU work that utilized representations learned from knowledge bases to enrich features of the current sentence, this paper directly learns a sentence representation incorporating memorized substructures with an automatically decided attention mechanism in an end-to-end manner.

## 3 Knowledge-Guided Structural Attention Networks (K-SAN)

Given an utterance with a sequence of words/tokens $\vec{s} = w_1, ..., w_T$, our NLU model is to predict corresponding semantic tags $\vec{y} = y_1, ..., y_T$ for each word/token by incorporating knowledge-guided structures. The proposed model is illustrated in Figure 2. The knowledge encoding module first leverages external knowledge to generate a linguistic structure for the utterance, where a discrete set of knowledge-guided substructures $\{x_i\}$ is encoded into a set of vector representations (§ 3.1). The model learns the representation for the whole sentence by paying different attention on the substructures (§ 3.2). Then the learned vector encoding the knowledge-guided structure is used for improving the semantic tagger (§ 4).

### 3.1 Knowledge Encoding Module

The prior knowledge obtained from external resources, such as dependency relations, knowledge bases, etc., provides richer information to help decide the semantic tags given an input utterance. This paper takes dependency relations as an example for knowledge encoding, and other structured relations can be applied in the same way. The input utterance is parsed by a dependency parser, and the substructures are built according to the paths from the root to all leaves (Chen and Manning, 2014). For example, the dependency parsing of the utterance "*show me the flights from seattle to san francisco*" is shown in Figure 3, where the associated substructures are obtained from the parsing tree for knowledge encoding. Here we do not utilize the dependency relation labels in the experiments for better generalization, because the labels may not be always available for different knowledge resources. Note that the number of substruc-

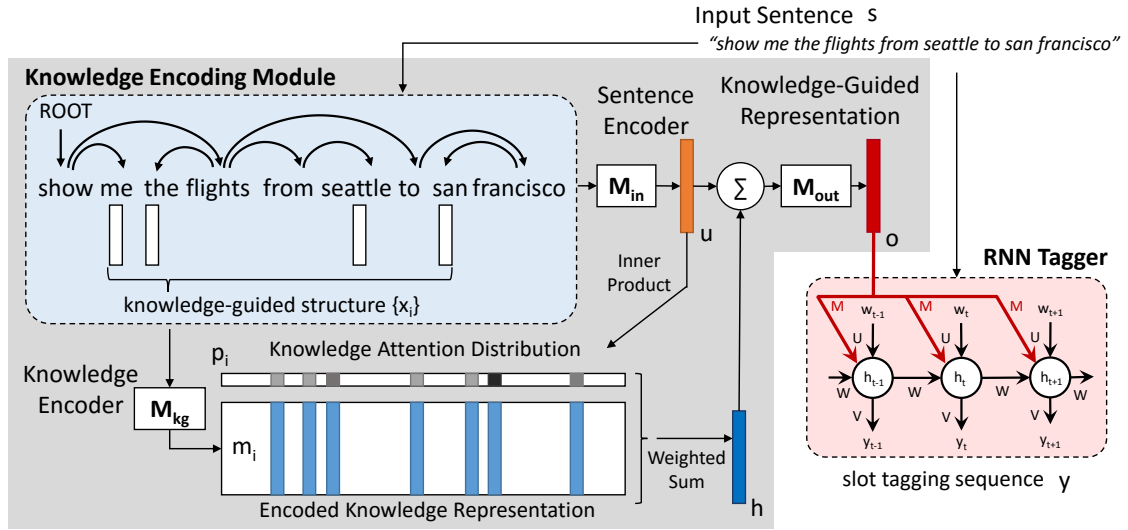

Figure 2: The illustration of knowledge-guided structural attention networks (K-SAN) for NLU.

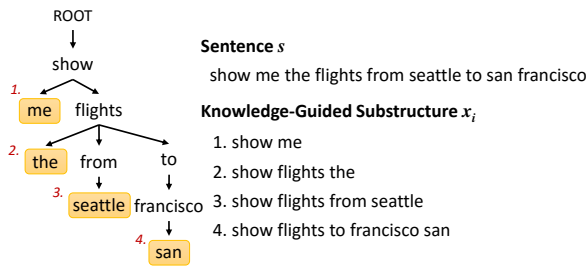

Figure 3: The knowledge-guided substructures of dependency parsing on an example sentence.

tures may be less than the number of words in the utterance, because non-leaf nodes do not have corresponding substructure in order to reduce the duplicated information in the model. The top-left component of Figure 2 illustrates the module for modeling knowledge-guided substructures.

### 3.2 Model Architecture

The model embeds all knowledge-guided substructures into a continuous space and stores embeddings of all $x$'s in the knowledge memory. The representation of the input utterance is then compared with encoded knowledge representations to integrate the carried structure guided by knowledge via an attention mechanism. Then the knowledge-guided representation of the sentence is taken together with the word sequence for estimating the semantic tags. Four main procedures are described below.

**Encoded Knowledge Representation** To store the knowledge-guided structure, we convert each substructure (e.g. path starting from the root to the leaf in the dependency tree), $x_i$, into a structure vector $m_i$ with dimension $d$ by embedding

the substructure in a continuous space through the knowledge encoding network $M_{kg}$. The input utterance $s$ is also embedded to a vector $u$ with the same dimension through the network model $M_{in}$. We apply the three types for knowledge encoding models, $M_{kg}$ and $M_{in}$, in order to model multiple words from a substructure $x_i$ or an input sentence $s$ into a vector representation: 1) fully-connected neural networks (NN) with linear activation, 2) recurrent neural networks (RNN), and 3) convolutional neural networks (CNN) with a window size 3 and a max-pooling operation. For example, one of substructures shown in Figure 3, "*show flights seattle from*", is encoded into a vector embedding. In the experiments, the weights of $M_{kg}$ and $M_{in}$ are tied together based on their consistent ability of sequence encoding.

**Knowledge Attention Distribution** In the embedding space, we compute the match between the current utterance vector $u$ and its substructure vector $m_i$ by taking their inner product followed by a softmax.

$$p_i = \text{softmax}(u^T m_i), \qquad (1)$$

where $p_i$ can be viewed as attention distribution for modeling important substructures from external knowledge in order to understand the current utterance.

**Sentence Representation** In order to encode the knowledge-guided structure, a vector $h$ is a sum over the encoded knowledge embeddings weighted by the attention distribution.

$$h = \sum_i p_i m_i, \qquad (2)$$

which indicates that the sentence pays different attention to different substructures guided from external knowledge. Because the function from input to output is smooth, we can easily compute gradients and back propagate through it. Then the sum of the substructure vector $h$ and the current input embedding $u$ are then passed through a neural network model $M_{out}$ to generate an output knowledge-guided representation $o$,

$$o = M_{out}(h + u), \qquad (3)$$

where we employ a fully-connected dense network for $M_{out}$.

**Sequence Tagging**    To estimate the tag sequence $\vec{y}$ corresponding to an input word sequence $\vec{s}$, we use an RNN module for training a slot tagger, where the knowledge-guided representation $o$ is fed into the input of the model in order to incorporate the structure information.

$$\vec{y} = \text{RNN}(o, \vec{s}) \qquad (4)$$

## 4   Recurrent Neural Network Tagger

### 4.1   Chain-Based RNN Tagger

Given $\vec{s} = w_1, ..., w_T$, the model is to predict $\vec{y} = y_1, ..., y_T$ where the tag $y_i$ is aligned with the word $w_i$. We use the Elman RNN architecture, consisting of an input layer, a hidden layer, and an output layer (Elman, 1990). The input, hidden and output layers consist of a set of neurons representing the input, hidden, and output at each time step $t$, $w_t$, $h_t$, and $y_t$, respectively.

$$\begin{aligned} h_t &= \phi(W w_t + U h_{t-1}), &(5) \\ \hat{y}_t &= \text{softmax}(V h_t), &(6) \end{aligned}$$

where $\phi$ is a smooth bounded function such as tanh, and $\hat{y}_t$ is the probability distribution over of semantic tags given the current hidden state $h_t$. The sequence probability can be formulated as

$$p(\vec{y} \mid \vec{s}) = \prod_i p(y_i \mid w_1, ..., w_i). \qquad (7)$$

The model can be trained using backpropagation to maximize the conditional likelihood of the training set labels.

To overcome the frequent vanishing gradients issue when modeling long-term dependencies, gated RNN was designed to use a more sophisticated activation function, such as long short-term memory (LSTM) and gated recurrent unit

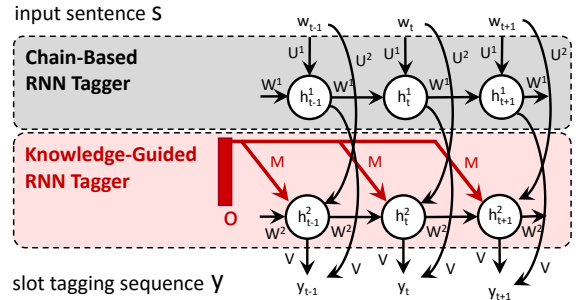

Figure 4: The joint tagging model that incorporates a chain-based RNN tagger (upper block) and a knowledge-guided RNN tagger (lower block).

(GRU) (Hochreiter and Schmidhuber, 1997; Cho et al., 2014). RNNs employing either of these recurrent units have been shown to perform well in tasks that require capturing long-term dependencies (Mesnil et al., 2015; Yao et al., 2014; Graves et al., 2013; Sutskever et al., 2014). This paper applies RNN with GRU cells to allow each recurrent unit to adaptively capture dependencies of different time scales (Cho et al., 2014; Chung et al., 2014), because RNN-GRU can yield comparable performance as RNN-LSTM with need of fewer parameters and less data for generalization (Chung et al., 2014; Jozefowicz et al., 2015).

### 4.2   Knowledge-Guided RNN Tagger

In order to model the encoded knowledge from previous turns, for each time step $t$, the knowledge-guided sentence representation $o$ in (3) is fed into the RNN model together with the word $w_t$. For the plain RNN, the hidden layer can be formulated as

$$h_t = \phi(Mo + W w_t + U h_{t-1}) \qquad (8)$$

to replace (5) as illustrated in the right block of Figure 2. RNN-GRU can incorporate the encoded knowledge in the similar way, where $Mo$ can be added into gating mechanisms for modeling contextual knowledge similarly.

### 4.3   Joint RNN Tagger

Because the chain-based tagger and the knowledge-guided tagger carry different information, the joint RNN tagger is proposed to balance the information between two model architectures. Figure 4 presents the architecture of

the joint RNN tagger.

$$h_t^1 = \phi(W^1 w_t + U^1 h_{t-1}), \quad (9)$$
$$h_t^2 = \phi(Mo + W^2 w_t + U^2 h_{t-1}), \quad (10)$$
$$\hat{y}_t = \text{softmax}(V(\alpha h_t^1 + (1-\alpha)h_t^2)), (11)$$

where $\alpha$ is the weight for balancing chain-based and knowledge-guided information. By jointly considering chain-based information ($h_t^1$) and knowledge-guided information ($h_t^2$), the joint RNN tagger is expected to achieve better generalization, and the performance may be less sensitive to poor structures from external knowledge. In the experiments, $\alpha$ is set to $0.5$ for balancing two sides. The objective of the proposed model is to maximize the sequence probability $p(\vec{y} \mid \vec{s})$ in (7), and the model can be trained in an end-to-end manner, where the error would be back-propagated through the whole architecture.

## 5 Experiments

### 5.1 Experimental Setup

The dataset for experiments is the benchmark ATIS corpus, which is extensively used by the NLU community (Mesnil et al., 2015). There are 4978 training utterances selected from Class A (context independent) in the ATIS-2 and ATIS-3, while there are 893 utterances selected from the ATIS-3 Nov93 and Dec94. In the experiments, we only use lexical features. In order to show the robustness to data scarcity, we conduct the experiments with 3 different sizes of training data (Small, Medium, and Large). The evaluation metrics for NLU is F-measure on the predicted slots.

For experiments with K-SAN, we parse all data with the Stanford dependency parser (Chen and Manning, 2014) and represent words as their embeddings trained on the in-domain data, where the parser is pre-trained on PTB. The loss function is cross-entropy, and the optimizer we use is adam with the default setting (Kingma and Ba, 2014), where the learning rate $\lambda = 0.001$, $\beta_1 = 0.9$, $\beta_2 = 0.999$, and $\epsilon = 1e^{-08}$. The maximum iteration for training our K-SAN models is set as 300. The dimensionality of input word embeddings is 100, and the hidden layer sizes are in $\{50, 100, 150\}$. The dropout rates are set as $\{0.25, 0.50\}$. All reported results are from the joint RNN tagger, and the hyperparameters are tuned in the dev set for all experiments.

### 5.2 Baseline

To validate the effectiveness of the proposed model, we compare the performance with the following baselines.

- Baseline:
  - CRF Tagger (Tur et al., 2010): predicts a semantic slot for each word with a context window (size = 5).
  - RNN Tagger (Mesnil et al., 2015): predicts a semantic slot for each word.
  - CNN Encoder-Tagger (Kim, 2014): tag semantic slots with consideration of sentence embeddings learned by a convolutional model.
- Structural: The NLU models utilize linguistic information when tagging slots, where DCNN and Tree-RNN are the state-of-the-art approaches for embedding sentences with linguistic structures.
  - CRF Tagger (Tur et al., 2010): predicts slots based on the lexical (5-word window) and syntactic (dependent head in the parsing tree) features.
  - DCNN (Ma et al., 2015a): predicts slots by incorporating sentence embeddings learned by a convolutional model with consideration of dependency tree structures.
  - Tree-RNN (Tai et al., 2015): predicts slots with sentence embeddings learned by an RNN model based on the tree structures of sentences.

### 5.3 Slot Filling Results

Table 1 shows the performance of slot filling on different size of training data, where there are three datasets (Small, Medium, and Large use 1/40, 1/10, and whole training data). For baselines (models without knowledge features), CNN Encoder-Tagger achieves the best performance on all datasets.

Among structural models (models with knowledge encoding), Tree-RNN Encoder-Tagger performs better for Small data but slightly worse than the DCNN Encoder-Tagger.

CNN (Kim, 2014) performs better compared to DCNN (Ma et al., 2015a) and Tree-RNN (Tai et al., 2015), even though CNN does not leverage external knowledge when encoding sentences. When comparing the NLU performance between baselines and other state-of-the-art structural mod-

| Model | | | | Dataset | | |
|---|---|---|---|---|---|---|
| | Encoder ($M_{kg}/M_{in}$) | Knowledge | Tagger | Small | Medium | Large |
| Baseline | - | ✗ | CRF | 58.94 | 78.74 | 89.73 |
| | - | ✗ | RNN | 68.58 | 84.55 | 92.97 |
| | CNN | ✗ | RNN | 73.57 | 85.52 | 93.88 |
| Structural | - | ✓ | CRF | 59.55 | 78.71 | 90.13 |
| | DCNN | ✓ | RNN | 70.24 | 83.80 | 93.25 |
| | Tree-RNN | ✓ | RNN | 73.50 | 83.92 | 92.28 |
| Proposed | K-SAN (NN) | ✓ | RNN | 74.11† | 85.97 | 93.98† |
| | K-SAN (RNN) | ✓ | RNN | 73.13 | 86.85† | **94.97**† |
| | K-SAN (CNN) | ✓ | RNN | **74.60**† | **87.99**† | 94.86† |

Table 1: The F1 scores of predicted slots on the different size of ATIS training examples, where K-SAN utilizes the dependency relations parsed from the Stanford parser. Small: 1/40 set; Medium: 1/10 set; Large: original set. († indicates that the performance is significantly better than all baseline models with $p < 0.05$ in the t-test.)

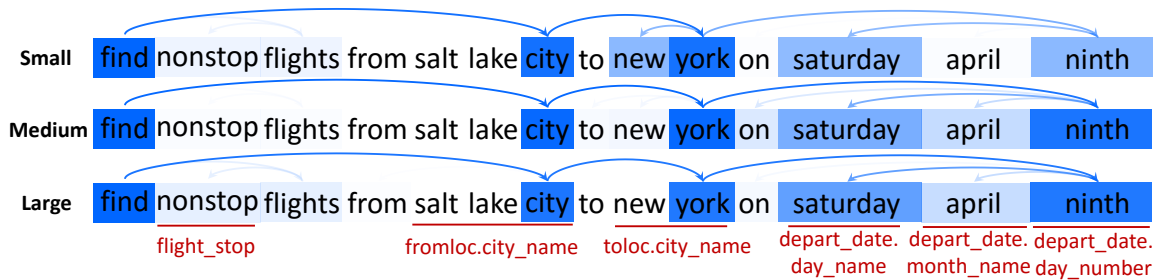

Figure 5: The visualization of the decoded knowledge-guided structural attention for both **relations** and **words** learned from different size of training data. Relations and words with darker color indicate higher attention weights generated by the proposed K-SAN with CNN. The slot tags are shown in the figure for reference. Note that the dependency relations are incorrectly parsed by the Stanford parser in this example, but our model is still able to benefit from the structural information.

els, there is no significant difference. This suggests that encoding sentence information without distinguishing substructure may not capture salient semantics in order to improve understanding performance.

Among the proposed K-SAN models, CNN for encoding performs best on Small (75% on F1) and Medium (88% on F1), and RNN for encoding performs best on the Large set (95% on F1). Also, most of the proposed models outperform all baselines, where the improvement for the small dataset is more significant. This suggests that the proposed models carry better generalization and are less sensitive to unseen data. For example, given an utterance "*which flights leave on monday from montreal and arrive in chicago in the morning*", "*morning*" can be correctly tagged with a semantic tag B-arrive_time.period_of_day by K-SAN, but it is incorrectly tagged with B-depart_time.period_of_day by baselines, because knowledge guides the model to pay correct atten-

tion to salient substructures. The proposed model presents the state-of-the-art performance on the large dataset, showing the effectiveness of leveraging knowledge-guided structures for learning embeddings that can be used for specific tasks and the robustness to data scarcity and mismatch.

## 5.4 Attention Analysis

In order to show the effectiveness of boosting performance by learning correct attention from much smaller training data through the proposed model, we present the visualization of the attention for both words and relations decoded by K-SAN with CNN in the Figure 5. The darker color of blocks and lines indicates the higher attention for words and relations respectively. From the figure, the words and the relations with higher attention are the most crucial parts for predicting correct slots, e.g. origin, destination, and time. Furthermore, the difference of attention distribution between three datasets is not significant; this suggests that

| Approach | Knowledge (Max #Substructure) | | | Small | Medium | Large |
|---|---|---|---|---|---|---|
| CRF | Syntax: Dependency Tree | Stanford | - | 59.55 | 78.71 | 90.13 |
| | | SyntaxNet | - | 61.09 | 78.87 | 90.92 |
| | Semantics: AMR Graph | Rule-Based | - | 59.55 | 79.15 | 89.97 |
| | | JAMR | - | 61.12 | 78.64 | 90.25 |
| K-SAN (CNN) | Syntax: Dependency Tree | Stanford | 53 | **74.60** | 87.99 | 94.86 |
| | | SyntaxNet | 25 | 74.35 | **88.40** | **95.00** |
| | Semantics: AMR Graph | Rule-Based | 19 | 74.32 | 88.14 | 94.85 |
| | | JAMR | 8 | 74.27 | 88.27 | 94.89 |

Table 2: The F1 scores of predicted slots with knowledge from different resources.

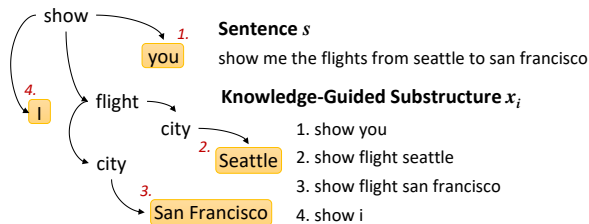

Figure 6: The knowledge-guided substructures of the AMR graph.

our proposed model is able to pay correct attention to important substructures guided by the external knowledge even the training data is scarce.

### 5.5 Knowledge Generalization

In order to show the capacity of generalization to different knowledge resources, we perform the K-SAN model for different knowledge bases. Below we compare two types of knowledge formats: dependency tree and Abstract Meaning Representation (AMR). AMR is a semantic formalism in which the meaning of a sentence is encoded as a rooted, directed, acyclic graph (Banarescu et al., 2013), where nodes represent concepts, and labeled directed edges represent the relations between two concepts. The formalism is based on propositional logic and neo-Davidsonian event representations (Parsons, 1990; Davidson, 1967). The semantic concepts in AMR were leveraged to benefit multiple NLP tasks (Liu et al., 2015). Unlike syntactic information from dependency trees, the AMR graph contains semantic information, which may offer more specific conceptual relations. Figure 6 shows an AMR graph associated with the example utterance as same as one in Figure 3 and how the knowledge-guided substructures are constructed.

Table 2 presents the performance of CRF and K-SAN with CNN taggers that utilize dependency relations and AMR edges as knowledge guidance on the same datasets, where CRF takes the head words from either dependency trees or AMR graphs as additional features and K-SAN incorporates knowledge-guided substructures. The dependency trees are obtained from the Stanford dependency parser or the SyntaxNet parser[1], and AMR graphs are generated by a rule-based AMR parser or JAMR[2].

Among four knowledge resources (different types and obtained from different parsers), all results show the similar performance for three sizes of datasets. The maximum number of substructures for the dependency tree is larger than the number in the AMR graph (53 and 25 v.s. 19 and 8), because syntax is more general and may provide richer cues for guiding more attention while semantics is more specific and may offer stronger guidance. In sum, the models applying four different resources achieve similar performance, and all significantly outperform the state-of-the-art NLU tagger, showing the effectiveness, generalization, and robustness of the proposed K-SAN model.

## 6 Conclusion

This paper proposes a novel model, knowledge-guided structural attention networks (K-SAN), that leverages prior knowledge as guidance to incorporate non-flat topologies and learn suitable attention for different substructures that are salient for specific tasks. The structured information can be captured from small training data, so the model has better generalization and robustness. The experiments show benefits and effectiveness of the proposed model on the language understanding task, where all knowledge-guided substructures captured by different resources help tagging performance, and the state-of-the-art performance is achieved on the ATIS benchmark dataset.

---

[1] https://github.com/tensorflow/models/tree/master/syntaxnet
[2] https://github.com/jflanigan/jamr

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
