# Peer review of "Knowledge as a Teacher: Knowledge-Guided Structural Attention Networks"

_ACL 2017 — decision unknown_

[Official Review · Reviewer 1 · rating 2 · confidence 4]
soundness 4 · originality 3 · clarity 4 · impact 3 · substance 4 · appropriateness 5 · meaningful comparison 2 · presentation format Poster

This paper proposes a neural network architecture that represent structural
linguistic knowledge in a memory network for sequence tagging tasks (in
particular, slot-filling of the natural language understanding unit in
conversation systems). Substructures (e.g. a node in the parse tree) is encoded
as a vector (a memory slot) and a weighted sum of the substructure embeddings
are fed in a RNN at each time step as additional context for labeling.

-----Strengths-----

I think the main contribution of this paper is a simple way to "flatten"
structured information to an array of vectors (the memory), which is then
connected to the tagger as additional knowledge. The idea is similar to
structured / syntax-based attention (i.e. attention over nodes from treeLSTM);
related work includes Zhao et al on textual entailment, Liu et al. on natural
language inference, and Eriguchi et al. for machine translation. The proposed
substructure encoder is similar to DCNN (Ma et al.): each node is embedded from
a sequence of ancestor words. The architecture does not look entirely novel,
but I kind of like the simple and practical approach compared to prior work.

-----Weaknesses-----

I'm not very convinced by the empirical results, mostly due to the lack of
details of the baselines. Comments below are ranked by decreasing importance.

-  The proposed model has two main parts: sentence embedding and substructure
embedding. In Table 1, the baseline models are TreeRNN and DCNN, they are
originally used for sentence embedding but one can easily take the
node/substructure embedding from them too. It's not clear how they are used to
compute the two parts.

- The model uses two RNNs: a chain-based one and a knowledge guided one. The
only difference in the knowledge-guided RNN is the addition of a "knowledge"
vector from the memory in the RNN input (Eqn 5 and 8). It seems completely
unnecessary to me to have separate weights for the two RNNs. The only advantage
of using two is an increase of model capacity, i.e. more parameters.
Furthermore, what are the hyper-parameters / size of the baseline neural
networks? They should have comparable numbers of parameters.

- I also think it is reasonable to include a baseline that just input
additional knowledge as features to the RNN, e.g. the head of each word, NER
results etc.

- Any comments / results on the model's sensitivity to parser errors?

Comments on the model:

- After computing the substructure embeddings, it seems very natural to compute
an attention over them at each word. Is there any reason to use a static
attention for all words? I guess as it is, the "knowledge" is acting more like
a filter to mark important words. Then it is reasonable to include the baseline
suggest above, i.e. input additional features.

- Since the weight on a word is computed by inner product of the sentence
embedding and the substructure embedding, and the two embeddings are computed
by the same RNN/CNN, doesn't it means nodes / phrases similar to the whole
sentence gets higher weights, i.e. all leaf nodes?

- The paper claims the model generalizes to different knowledge but I think the
substructure has to be represented as a sequence of words, e.g. it doesn't seem
straightforward for me to use constituent parse as knowledge here.

Finally, I'm hesitating to call it "knowledge". This is misleading as usually
it is used to refer to world / external knowledge such as a knowledge base of
entities, whereas here it is really just syntax, or arguably semantics if AMR
parsing is used.

-----General Discussion-----

This paper proposes a practical model which seems working well on one dataset,
but the main ideas are not very novel (see comments in Strengths). I think as
an ACL paper there should be more takeaways. More importantly, the experiments
are not convincing as it is presented now. Will need some clarification to
better judge the results.

-----Post-rebuttal-----

The authors did not address my main concern, which is whether the baselines
(e.g. TreeRNN) are used to compute substructure embeddings independent of the
sentence embedding and the joint tagger. Another major concern is the use of
two separate RNNs which gives the proposed model more parameters than the
baselines. Therefore I'm not changing my scores.